# Understanding the Diversity of People in Sex Work: Views from Leaders in Sex Worker Organizations

**Andrea Mellor [1,*] and Cecilia Benoit [1,2]**

[1]   Canadian Institute for Substance Use Research, University of Victoria, Victoria, BC V8P 5C2, Canada
[2]   Department of Sociology, University of Victoria, Victoria, BC V8W 2Y2, Canada
[*]   Correspondence: amellor@uvic.ca

**Abstract:** Criminal laws in Canada and many other countries are currently premised on the assumption of homogeneity, that is, people in sex work are cis women and girls who are being sexually exploited/sex trafficked. This perspective is also shared by antiprostitution groups and many researchers investigating the "prostitution problem". Perpetuating this position obscures their demographic multiplicity and variety of lived experiences. We interviewed 10 leaders from seven sex worker organizations (SWOs) across Canada who reported a diversity among their clientele that is rarely captured in the extant literature and absent from the current Canadian criminal code. Our findings reveal the important role that SWOs have to play in facilitating access to health and social services and providing spaces where people in sex work can gather in safe and supportive environments, without the fear of stigma, discrimination, or police harassment. We conclude that SWOs can operate as a structural intervention beyond decriminalization that can improve equitable access to health and social services for sex workers Despite SWOs' efforts, sex workers' mobilization is still limited by micro-, meso-, and macrolevel stigmatization that prevents and/or discourages some workers from accessing their programs and services.

**Keywords:** sex work; structural interventions; stigma; occupational health; diversity; intersectionality

## 1. Introduction

Much of recent government policy in the Global North is based on the assumption that commercial sex relationships are, by nature, unequal: female sellers are forced to engage, and male buyers are in control of the interaction. The Swedish prostitution policy is a case in point, as it understands prostitution as patriarchal oppression (Östergren 2017) and "sex-selling females as victims of 'men's violence against women'" (Florin 2012, p. 217). Several European countries have embraced the "end-demand" policy approach involving the banning of sexual purchase and most other prostitution-related activities (sometimes referred to as the "Swedish model" or "Nordic model") (Benoit et al. 2019). This includes Norway and Iceland in 2009, Northern Ireland in 2015, France in 2016, and the Republic of Ireland in 2017 (Harrington 2017). Canada has followed suit, whereby its 2014 Bill C-36 contends, "prostitution [is] a form of sexual exploitation that disproportionately and negatively impacts women and girls" (Department of Justice Canada 2014).

This illusion of sex worker homogeneity is found among antiprostitution lobby groups and academic researchers who argue that sex work is principally an institution of hierarchal gender relations that legitimizes the sexual exploitation of women by men (Coy et al. 2019; Coy et al. 2012; Farley 2018; Moran and Farley 2019). Many researchers providing supportive empirical evidence to the decriminalization of sex work also restrict their sampling to cis and/or trans women (e.g., Argento et al. 2020; Harcourt et al. 2010; Lazarus et al. 2012; Machat et al. 2019; McBride et al. 2019; West et al. 2022). A recently published study (Bungay et al. 2023) examined 64 research projects funded by Canada's premier health research funding agency, the Canadian Institutes for Health Research, in the past

two decades and found that there was a major focus on street-based sex work, limited discussion of sociostructural contexts the sex industry, and a perpetuation of stigmatizing narratives about sex work. When gender is noted in the study abstracts, all included women, with only two also including men and transgender people. Most grants focused on the sexual practices and sexual infections among sex workers or injection drug use. Data on racial identities were not listed. The authors conclude that "[r]esearch on indoor sex work and virtual sex work, along with the complexity of actors involved is largely missing" (p. 77).

The simplistic framing underpinning criminal code laws and much of the current sex work research is starkly different from the reality faced daily by community organizations providing health and social services and social support to people in sex work in their communities (Bungay et al. 2012, 2023; Jiao and Bungay 2019; Shaver 2018). In contributing to this Special Issue, our objective is to investigate the diversity among the clientele of social organizations serving sex workers, which we refer to here as sex worker organizations (SWOs). These "underground organizations" (Anasti 2017, p. 418) in many ways operate as structural interventions that are largely invisible in the formal health and social service system (Benoit et al. 2021a).

Two key roles of SWOs are reducing the social isolation of the spectrum of people who sell sexual services and bringing them to available services and programs (Argento et al. 2011; Azhar et al. 2020; Vijayakumar et al. 2019). Access to health and social services for people in sex work in their communities is challenged by the stigma of their work; their disadvantaged social locations, which are due to poverty, racial, sexual minority, and disability statuses; their economic marginalization; and the criminalized frameworks that surround their occupation (Benoit et al. 2018; Benoit and Unsworth 2022; Foley 2019; Vijayakumar et al. 2019). The role of SWOs in supporting the inclusion of sex workers from diverse backgrounds has been shown to effectively address social and health inequities, improve quality of life, and enhance sex workers' wellbeing (Belle-Isle et al. 2014; Benoit and Unsworth 2022; Swendeman et al. 2015; West et al. 2021).

However, as we investigate below, SWOs are challenged by society's hierarchical order system, which ranks people in sex work from "privileged" to "lesser" (Sciortino 2016; Witt 2020). This "whore-archy" detracts attention from the most marginalized workers and strengthens horizontal stigmatizing attitudes within the sex worker community, preventing access to supports (Graceyswer 2020). This social stratification extends beyond the realm of sex work policy (Vijayakumar et al. 2019) and may replicate and/or strengthen inequities that already marginalize and isolate workers (Fuentes 2022; Kotiswaran 2011).

To shed light on this complex issue, we present findings from leaders of seven SWOs across Canada. Participants described the demographic makeup of their clients, a variety of ways that they navigate the challenges of working with and accommodating a diverse clientele, how they struggle to provide support and outreach to marginalized sex workers in their community, and ways they build leadership capacity among their clients to support their engagement at multiple levels. This includes advocating for rights, evidence-based policy, and equitable access to health and social services. Although our participants shared ways that they engaged with the sex worker community at multiple levels, they also discussed how their reach extends only as far as their networking capabilities. While SWOs operate as an important resource for access to a variety of health and social service resources, challenges with "being out", horizontal discrimination among workers, and other complexities associated with social exclusion beyond those imposed by policy measures continue to challenge SWOs aiming to mobilize services to all diverse corners of the sex industry.

## 2. Integrated Conceptual Framework

In this paper, we employ a combined intersectional and life course framework to shed light on the diversity of sex workers. Intersectionality, both a theory and a method, focuses on differences among groups and illuminates various and interacting social factors

that may affect individual lives, including social location, health status, and quality of life (Hankivsky 2005; Hankivsky and Christoffersen 2008). Sex and gender intersect with Indigeneity, social class, race, and other influences that collectively cause varying health and social outcomes across time and place, including how individuals access protective and healthcare services and how they are treated by service providers (Hawkes and Buse 2020).

Theoretically, intersectionality posits that various dimensions of social stratification, including sex and gender but also socioeconomic status, ethnicity, race, and so on, can add up to or culminate in greater disadvantage for some people while privileging others (Benoit and Unsworth 2022). Importantly, this approach exposes the complexities encountered when aligning with "feminism governance": the installation of feminists and feminist ideas in the actual legal-institutional power (Halley et al. 2006). Despite its important role in drawing attention to, for example, wartime sexual violence, feminist governance has been co-opted by numerous governments and conservative feminist organizations to justify legislative frameworks that criminalize sex work as a measure to protect vulnerable women and girls (Halley et al. 2006; Kotiswaran 2011). Intersectionality shows how sex and gender intersect with Indigeneity, social class, race, and other influences that collectively cause varying health and social outcomes across time and place, including how individuals access protective and healthcare services and how they are treated by service providers (Hawkes and Buse 2020). Therefore, intersectionality problematizes the biases invisibilized by feminist governance, namely the stigmatization experienced across multiple social inequalities (Benoit et al. 2019; Foley 2019; Kotiswaran 2011; McMillan and Worth 2019; Vijayakumar et al. 2019).

In practice, an intersectional perspective understands that an individual's lived and living experiences are produced by multiple social positions (e.g., age, race, immigrant status, gender, class, ability, sexuality, etc.) and cannot be effectively understood by examining these social factors separately (Bauer et al. 2021). Therefore, intersectionality can access the complex lived realities of individuals across societal systems and structures and how this reality is malleable and circumstantial, depending on individuals' relationships with others around them (Iyer et al. 2008). The life course framework provides a temporal complement to intersectionality, recognizing that social locations are influenced by cohorts, transitions, trajectories, life events, and turning points and, importantly, that experiences earlier in life can have long-term consequences (Belle-Isle et al. 2014; Benoit et al. 2021a; Hutchison 2005).

### 3. Materials and Methods

#### 3.1. The Study

The research results reported below are part of a larger participatory research project (Benoit and Unsworth 2021, 2022) led by the second author that focused on reducing the isolation of people in sex work in their communities. The project aimed to find ways to equip a new generation of sex workers with the tools to respond to negative messaging about their occupation and mobilizing them to advocate for their human rights, including safe working conditions, dignity, respect, and social inclusion in Canadian society. One concrete outcome of the project was the development of an intervention strategy to reduce the isolation of people in sex work and bring them to outreach services and community gatherings. The bilingual (English and French) mobilization tool has been, to date, piloted in five communities.

#### 3.2. Interview Procedure

Qualitative data were collected from in-person interviews conducted in November 2020 with 10 cisgender female staff members (henceforth referred to as "participants") from seven SWOs located in different communities across Canada, ranging from large to small cities.

Most of the SWOs are peer driven, providing support, advocacy, and education by, for, and with sex workers who currently work, or who formerly worked, in the sex industry.

Some SWOs only cater to women-identified (cis and trans) sex workers, whereas others provide services to all genders. The SWOs also vary in their emphasis on service provision versus advocacy. Programs include primary healthcare, street outreach, the delivery of safe sex and drug use supplies, peer drop-in, weekly community meals, occupational health and safety, violence prevention workshops, public education, and research. Some SWOs have been in existence for decades and have been incorporated as nonprofits with a valid charity number from the Canada Revenue Agency. Others do not have charitable status and do not qualify as legitimate recipients of government funds. A few SWOs have stable multiyear funding; most do not, requiring staff to apply on an ongoing basis for grants to fund programs. The grants that are available tend to be targeted toward gender-based violence supports and services for cis women, HIV, drug overdose, harm reduction and prevention, and, to a lesser extent, housing.

Of the interviewees, 7 out of 10 have lived and living sex work experience; all have extensive experience working on the frontline of service delivery to sex workers and advocating for their human rights. Participants' job descriptions include executive director, operations manager, violence prevention coordinator, team leader for prevention and support, program coordinator, advocacy coordinator, and coordinator of communications and mobilization. Three participants were from one SWO, two participants were from another SWO, and five participants were from each of the other five SWOs. Qualitative data were collected from in-person and telephone interviews. Verbal informed consent was obtained from all participants, and permission was granted for the use of audio-recording equipment during the interview. Participants were assured of their ability to end the interview at any time and of the confidentiality of the data that they shared with researchers. Ethics approval for this portion of the study was obtained from the authors' institution.

The interview schedule consisted of 12 open-ended questions, codeveloped with the research team. During team teleconference calls, it was agreed that the second author would create an initial draft of these research questions, and participants were invited to review and give feedback on the draft questions. All did so and made recommendations to ensure the questions were not personalized but rather were focused on their SWOs and the developing research project. For this reason, limited descriptive data are provided of the participants themselves at their request, as they instead are designated to speak on behalf of their respective SWO rather than their own experiences or affiliations with the sex industry.

Interviews, ranging in length from 30 to 60 min, were all conducted by the second author. To acknowledge their time, knowledge, and expertise, participants received an honorarium of CAD 100, which some said they would donate to their SWO. All audio recordings were transcribed, and all identifying details were redacted. The transcribed interviews were then sent to participants to check for accuracy. A few participants requested the removal of small passages of text that could have identified their SWO.

### 3.3. Thematic Analysis

Below, we present our analysis of participants' answers to a subset of questions that highlight the multiple types of diversity among sex workers in their communities, tensions that arise because of diversity, and ways that their SWOs manage these tensions to establish better organizational and professional practice: (1) How has your group accounted for the inclusion and leadership building of Indigenous, Black, Asian, migrant sex workers, sex workers who use drugs, and youth involved in sex work? Who else should be mobilized? How do you handle in-group tensions? (2) How is participants' being out, not being out, or needing to maintain a low profile going to impact the mobilization that is possible for this project? What are some of the ways that your organization has thought through this issue, and how does it address the issue?

Participants' transcribed answers were coded by using NVivo 10 software (QSR International Pty Ltd., Burlington, VT, USA), following Braun and Clarke's (2006) multistep approach to thematic analysis. An abductive approach was used to guide the transcript

review, identifying themes on the basis of probes specific to the research question and "emergent themes" related to the research questions but not probed by the interviewer. An abductive approach moves between pre-existing knowledge and the data to identify new knowledge, insights, and novel findings in the research (Kolko 2010; Thornberg 2012).

The authors read all the transcripts multiple times to gain familiarity with the data, and two transcripts were selected for the authors to independently code them to reach consensus on the coding structure before applying it to the full data set. The thematic interpretation of the data was then organized into master themes (i.e., intersections of diversity in sex work, creating safe spaces for diversity at the local level, and mobilizing the diversity of sex workers at multiple levels). Theoretical insights from the perspectives of intersectionality and sex work (Bauer 2014; Foley 2019; Hankivsky and Christoffersen 2008; McMillan and Worth 2019; Vijayakumar et al. 2019), the sociology of stigma (Goffman 1963; Hatzenbuehler et al. 2013; Link and Phelan 2014), and capabilities (Nussbaum 2003) were employed during our analysis of the participants' detailed responses to questions about how to mobilize diversity among sex workers and organizations and about constraints faced in doing so. These verification techniques were employed to help increase rigor in the qualitative analysis and interpretation.

### 3.4. Locating Ourselves

The first author is a postdoctoral fellow whose participatory research practice focuses on improving access to social, cultural, and environmental determinants of health on the basis of the lived and living experience of diverse community stakeholders. The second author researches barriers to health, safety, and rights for people who sell sexual services, among other marginalized populations. She adopts a comprehensive community-based participatory approach to find diverse samples of participants to answer her research questions.

## 4. Findings

Below, we report participants' answers to two related questions about how their SWOs account for inclusivity across dimensions of diversity and address challenges that arise. All participants answered both questions with some detail. We organized our codes into three emergent themes: (1) intersections of diversity in sex work, (2) creating safe spaces for mobilizing diversity at the community level, and (3) mobilizing the diversity of sex workers at multiple organizational levels.

### 4.1. Intersections of Diversity of People in Sex Work

Participants discussed the various ways that diversity is expressed across their SWOs and how this influences how they tailor mobilization strategies to reflect their clientele. The range among clients included intersecting social demographic factors, including gender, sexuality, Indigeneity, race, social class, and age, as well as other distinctions/markers, such as (dis)ability, diverse worldviews, particularly Indigenous ways of knowing and doing, working conditions, living conditions, life stages such as using drugs and/or parenting, and age when clients began sex work. As one participant indicated, "in our work, it doesn't come down to the diversity of sex work experience so much as the diversity of people". By exploring the multiple dimensions of diversity within SWOs, they can more effectively, as one participant put it, "[speak] to different realities".

A range of genders within the sex worker community was discussed by many participants, including people in sex work identifying as cis women, cis men, trans women, trans men, gender nonbinary, and those who have "experience working as women, even if that's not their gender identity". Some participants shared that representation from trans women and nonbinary workers had increased in recent years, causing more programs to be offered to support their needs.

While a range of genders was present in all SWOs, representation among diverse groups varied. One participant shared about the unequal representation of cis women and cis men at one of their meetings,

> "The two cis men who were in the group were feeling really ostracized, or under-represented. Which, there's a certain irony in that. But yeah, they didn't feel entirely comfortable in the group even though we had some, I think, some good group guidelines in place, but they were just feeling like kind of outnumbered. ... And like, we didn't have any nonbinary, or Two-Spirit folks, it was all like cis men, cis women."

Another participant shared that male-identified members presently comprised only about 5% of their clientele: we would like to "hold some, like, community consultations and put out a little survey to try and figure out how we can be more welcoming and safer for folks on that end of the spectrum".

While creating spaces of safety for members is crucial, as a consequence, it privileges certain genders and marginalizes others. One participant related the following:

> "We'll work with men who have worked as women. But we don't work specifically with cisgender men, who work as men. ... Ah, you know, you know, if men wanted to organize around their own mobilization in sex work, then, you know, we would like to support that. But we need to, for the sake of the safety and the work of [SWO], and our resources, we need to keep it centered on women, and, ah, trans women."

Despite this current practice, the same participant expressed that their SWO wanted to be more age and gender inclusive: "I think youth is one that we really want to make space for, and make sure they have support in terms of mobilizing and connecting with our mobilization. And also men. Men in the sex industry".

All the participants either talked about how their SWO was prioritizing Indigenous engagement and/or had enhanced their programming to accommodate Indigenous people in sex work. Most participants referred to Indigenous women, and one participant discussed Two-Spirit[1] identities. Two participants mentioned that a substantial minority of their clientele identified as Indigenous:

> "We just did our strategic planning, and that was one of the top things that came out, was the desire to foster more Indigenous leadership within the organization. Recognizing that statistically, the folks that are accessing services at [SWO], um, you know—I can't remember the exact number, but maybe it's around forty percent or something, identify as Indigenous. So just wanting to have more staff, more peer-leadership, more, um, you know, recognizing. [L]ike even in terms of some of our intake forms, like asking people for example where they're from, to talk a little bit about who they are, not just are you Indigenous or not. Like that kind of thing; like just having more conversations as opposed to checklists. ... [And] of course doing territorial acknowledgments, and trying to just create a more respectful space."

The absence of diverse clientele/leadership from other marginalized communities was attributed by some to broader structural histories of oppression and social class distinctions. One participant recognized that while there was a variety of services available for transgender sex workers, their representation in leadership roles was limited. Another participant shared that "Black, Asian, and migrant communities of sex workers are truly not as visible or prominent in this place [city, province]". Another participant concurred: "I can't really say that we have super strong leadership from Black and migrant and Asian communities, because of, well, the isolation, and years of colonization, and our social culture".

Despite the desire of SWOs to be inclusive, one participant suggested that *classism*, a belief that a person's social or economic station in society determines their value in that

society (Merriam-Webster 2023), was a systemic barrier. In particular, they suggested that this persisted between Canadian-born and im/migrant sex workers:

> "I think especially in Canada, [members of] the sex worker movement . . . they have like more education background, and more experience of involving mobilization, or lobbying, or advocacy, or having the organizational experience. I think it is important to mobilize the grassroots sex workers that may not have that access of language or access of like, the information and knowledge."

Another participant observed that the sex worker movement in their province has "historically [been] a very white movement, a very socialist movement, with . . . values that didn't necessarily account for different diversities across race in particular. Across class more so, but, not across race necessarily or other positionalities".

Differences between clients' social class backgrounds also highlighted diverse reasons for engaging in sex work and intersections with the "whore-archy". When piloting the mobilization tool in their local community, one participant shared how class became visible during the workshop:

> "There was a couple of people in the room who don't have fancy educations, and who live in the [downtown], and then maybe another three or four people who don't live in the [downtown], are in a way higher tax bracket, have a couple of jobs, and they do sex work on the side."

This same participant then shared, "people [say] things like, 'I'm not comfortable hanging out in the drop-in space. There're people who smell bad and who do drugs there, and I'm not okay with that'".

Another participant related:

> "We have women in the group who are homeless and are sleeping in the bush and coming in to get to meetings. [T]hey're not coming into the meeting showered and put together in the same way that someone who's securely housed and has access to running water and all those things is able to show up."

Another participant expanded on ways that some clients may face other challenges in their lives, highlighting additional dimensions of diversity that organizations need to attend to:

> "Living with HIV or hep C . . . it's Indigenous women, it's women who are using or have used drugs. It's ah . . . you know, women who have mental health challenges, that you know, are trying to work through that. And then, you know, women who are precariously housed. And these are all, these are really the key issues in our community."

Another participant spoke about the underrepresentation of "the intersections with sex workers and folks with disabilities, or chronic illness, especially invisible illness and things that aren't physically demobilizing . . . like mental health and, yah, just chronic invisible illness".

Other sites of exclusion included the accessibility of language resources at SWOs, which might limit how much people in sex work who have English as a second language can engage in programming and mobilization activities. Conversely, for English-speaking sex workers working in French-speaking provinces, the lack of English-translated materials was noted as a challenge. One participant said about language, "I hadn't considered that as an English-speaking sex worker in a French-speaking province that that was [a] basis for . . . isolation".

When discussing clients who were using drugs, one participant said they ask clients for tolerance and to be accepting of each other's diverse experiences with substance use: "Maybe, you're not there now, but maybe you were at one point? And just trying to view their journey through that lens". Many clients of SWOs also had shared histories, with one participant saying,

> "Sometimes there's tensions around stuff that's happened outside of [SWO]. Personal relationships among the group. Because some of the women, or many of the women in the group, have known each other for, since they were teenagers. And, you know, the majority of the women in the group are in their late thirties, early forties, or fifty even. Like . . . there's lots of history there. Lots of really complex challenging history."

Because of the potential for challenges to arise outside of the organizations, participants talked about the importance of self-assessment based on the diverse life stages of clients. This included the need for each individual to weigh the risks and benefits to being out within the context of their present-day personal lives and relationships. One participant shared, "a lot of us in the group, our kids are grown, and that makes a big difference about being out or not, when we don't have the potential of . . . you know, losing the right to parent your children".

*4.2. Creating Safe Spaces for Mobilizing Diversity at the Community Level*

Participants discussed their SWOs' strategies to create "safe spaces" of engagement as an essential mechanism in accommodating the diversity of their clients. Strategies included creating organizational guidelines on the basis of compassionate values, having special interest groups for clients from similar social or demographic locations, having workshops among clients to collaboratively determine how to cultivate inclusive spaces, and being accommodating to the spectrum of disclosure among clients (e.g., out to all, out to some, or out to none). The strategies of client engagement responded to unique geographies, histories, and demographic contexts of the SWOs, but all aligned in their collective awareness of the importance of cultivating safe spaces to reach the greatest number of clients possible and enhance access to low-barrier and nonjudgmental services.

Guidelines presented by participants were grounded in organizational values. For one SWO, this included "values around antiracism, around trans-inclusion, that migrant rights and immigration are things that we need to be aware of and continue to support". Strategies also mentioned using nondiscriminatory language, creating welcoming spaces, and being, as this participant related, "inherently LGBTQ inclusive". Another participant said, while "finding and creating safety for everybody" could be a challenge, maintaining alignment with philosophies such as "by and for" and "welcoming diversity" helped. This participant put it the following way:

> "Just as a kind of balancing point around respect and, and, and um, respecting diversity, and recognizing diversity . . . exploring and sharing and creating space for people to share that, even in the different kinds of sex work people are doing. Whether they're doing street level, or what some people might call survival sex verses . . . folks . . . may be doing escorting or different kinds of sex work. . . . [J]ust making space for people to share where they're coming from and trying to kind of break down those kind of stigmas within sex work."

Another participant narrated:

> "If folks haven't experienced homelessness, or they haven't experienced dire poverty, I can understand why coming down here can be, like, difficult. But at the same time, if people are coming here to access our services, but they think that they are somehow better than the people who live in the [downtown] or who work outdoors, then they're not safe to be in that space, because they are going to harm our other members."

Participants discussed the need for organizational reflexivity on the question "how are we making ourselves safe enough?" in guiding the types of services and the programming they developed, such as ensuring that clients, as one practitioner said, "feel honored, feel not discriminated against, um, you know, talking about not outing our community members and stuff like that". Another participant shared:

"We've had special groups that have come together to create space, safe space for specific populations. So, for example, a trans/nonbinary group, an Indigenous group, a men's program and group, indoor workers group. And um, you know also doing um . . . really trying to work with the Indigenous agencies and nations to bring in some more Indigenous ways of knowing."

When challenges related to language arose within an SWO that supports people in sex work and people living positive with HIV, one participant related,

"We asked the community, 'what's the best way that we should tackle these tensions?' And some of our community advisory members said that they would like to see a workshop hosted about language, and how to use inclusive language. And then we actually had a pretty good turnout. We had about half sex workers and half . . . people living with HIV attend our workshop, and we had a discussion about inclusive language with both groups. And we used that information to inform some of the language that we use here in our space. And then we also used that, actually then piggy-backed off, we started having peer-led workshops on language after that."

Cultivating safe spaces of engagement to accommodate diverse lived experiences across the life course also required flexibility and understanding around peoples' ability and/or desire to be known as working in the sex industry. Participants indicated that "being out" was relative and depended on the social circumstance, the location, and the time in that person's life. For example, some workers may be "out" among their peers, but not socially with their friends and family. As one participant put it, just because "some people feel comfortable to share in certain circle[s] doesn't mean they feel comfortable to share with the public or law enforcement".

*4.3. Mobilizing the Diversity of Sex Workers at Multiple Organizational Levels*

Finally, participants discussed how their SWOs have striven to accommodate client strengths and the capabilities of people in sex work to "meet them where they are at". At the leadership level, some participants recognized that organizational structures need to be disrupted, by enacting equity-centered hiring practices and learning ways that individual social locations can be leveraged to contribute to mobilization efforts. Recognizing that clients are experts of their own lived experiences and that there is power in connecting across those experiences, participants discussed ways this can be an invaluable tool for community outreach, mobilization, and strengthening organizations' abilities to respond to on-the-ground issues. All participants agreed that a variety of leaders was needed to mobilize the diversity of people in sex work. Different strategies were suggested, including embedding equity in their organizations' hiring practices to ensure that the voices of sex workers across the industry were represented in decision-making and strategic planning. As one participant put it, "we see all sex workers especially racialized sex workers, drug using sex workers, as experts of their own experience, and try to always find ways for them to guide parts of our collective process around equity".

Participants also stated that their SWOs need to "qualify" equity in terms of their own diverse clientele and use those metrics to develop initiatives and evaluate progress. For one participant's SWO, this included "mobilizing people who work on the street into power positions". Others said this required revising "the hiring practices . . . for how we build leadership and inclusion of Indigenous, Black, Asian, migrant sex workers, and sex workers who use drugs", and "in terms of the leadership building . . . we're actively working right now to create positions of leadership for Indigenous women that have a lived experience in sex work". Hiring sex workers from marginalized communities "really facilitates that inclusion and facilitates that leadership building because it's how we met other sex workers . . . who identify in these ways".

Some SWOs hosted peer-led groups or workshops to discuss how to build diverse leadership capacity. Potential topics included addressing tensions within the sex worker

community (e.g., "whore-archies", classism, etc.), building capacities around trauma-informed counseling, communication techniques, harm reduction, and how to be social justice mentors. Such training initiatives were deemed important in setting clients up for success, as one participant reflected:

> "How can we have entry points to doing this work and becoming involved, that reflect where people are at, without launching [them] prematurely into employment situations where they don't do well, and then it becomes another source of stress or failure in [their] life?"

Participants also spoke about committees identifying targeted outreach opportunities where peers could go into their communities and report back on issues of importance that SWOs may not yet be addressing. Exercising the "by and for" philosophy, one participant related,

> "How we offer services, and how we articulate our politic is based on what happens on the ground. [A]nd 'by the ground,' I just mean like the outreach team comes back and says, 'Okay, this is what's needed,' and then we respond to that".

While participants were cognizant of the need to be inclusive of the diversity in their communities, they acknowledged the presence of structural barriers within their SWOs that prevented the most marginalized from holding leadership positions. As one participant stated,

> "I think what tends to happen, though, is positions of power in organizations tend not to be led by people of more marginalized groups, and that's because organizations are . . . organized in ways that maybe don't represent, they elect, or, sort of like, a colonized model?"

In addressing this accessibility barrier, participants discussed ways that clients needed support and mentorship in learning how to safely leverage their unique positions, be it socioeconomic privilege or connections to harder-to-reach groups, all of which will contribute to their gaining strength in joining mobilization activities and having success in leadership roles. As one participant shared,

> "There are so many different options of engagement, so many different ways to tap in. . . . That there can be art, there can be social media, there can be written components, research components. We make sure to offer *so* many options, even descriptively, that do not require your name and your face to be attached".

Another participant communicated a similar perspective, paying special attention to their clients' decisions to be "out or not out":

> "We have created a working group that holds meaningful contribution, um, regardless of whether you want to be out on stage and identify yourself as someone who has lived experience, or out on stage and identify as an ally, or, you know, just at the table doing art and contributing to the conversation in that way or showing up and identifying as an ally."

As another participant related,

> "We are giving different ways people can identity. So people can come [and] identify as a volunteer, they can identify as a worker, they can identify they work in [a] massage parlor, they can identify they are [a] sex worker."

Participants discussed the importance of "trying to find more creative ways to get those voices out there" because "[w]hen there's drug users on the team, it's easier to mobilize sex workers who are drug users. When there's Indigenous women on the team, it's easier to mobilize Indigenous sex workers. Um, because people like to see themselves in the organization and the staff represents the organization".

Because the capacity for different groups to access spaces of influence was identified as a challenge, it was necessary for some SWOs to reach outside of the sex worker community

to identify community allies. Being strategic about having limited resources meant, for some participants, "instead of going out sort of like broadly and speaking to everybody, [asking instead] who are the stakeholders you have to meet, and how can you bring them to you to hear about things in maybe a more controlled environment".

This was also a strategy for supporting groups with greater complexities, specifically supporting sex workers who were minors. While some participants talked about working with youth-serving organizations and/or partnering with other community programs that also addressed those issues, SWOs are largely constrained by the legal implications of supporting youth engaged in sex work. The underlying moral debate was explained by one participant as follows:

> "There's huge tension around whether sex work is something that is understood as work and a decision people make, or if it's only understood as exploitation and violence. So working with youth often is, working with people under eighteen is, often very difficult for, I think, sex worker organizations because there's not a consensus on what under-eighteen experiences are like. What they need. What that advocacy should look like. So we really rely on connecting with people who had experiences under eighteen who are now over eighteen and can share with more agency and more awareness, what they would have needed, what that was like for them."

In the end, to be strategic about mobilization, SWOs needed opportunities to reflect on which activities were successful and which were less effective. According to the following participant, diversity within and across SWOs is always evolving and changing:

> "We've existed for twenty-five years now, and there have been times when the makeup of the organization itself is not diverse in any way, really. With regard to race or class or work positions. And sometimes extremely diverse with regard to all of those things."

## 5. Discussion

Currently, the criminal laws surrounding sex work in Canada and many other countries are premised on the assumption of homogeneity, that is, that people in sex work are cis women and girls who are being sexually exploited/trafficked (Coy et al. 2011; Coy 2012; Farley et al. 2004; Farley 2004, 2007; Kelly et al. 2009; McBride 2020; Poulin 2008). This perspective is also shared by antiprostitution groups and many researchers investigating the "prostitution problem" who ignore or overlook the diversity within the sex industry (Coy et al. 2019; Farley 2018; Farley et al. 2005). Even activists and researchers who call for the decriminalization of sex work and provide supportive empirical evidence have at times restricted their sampling to cis and/or trans women (e.g., Argento et al. 2020; Harcourt et al. 2010; Lazarus et al. 2012; Machat et al. 2019; McBride et al. 2019; West et al. 2022). Moreover, within the industry, horizontal discrimination exists that marginalizes certain factions of workers, an occupational stratification referred to as the "whore-archy" (Graceyswer 2020; Sciortino 2016; Witt 2020). The result of limiting inclusivity across these dimensions is the continued oppression and stigmatization of sex workers, leading to their exclusion from equitably accessing comprehensive health, protection, and social services (Argento et al. 2020; Benoit et al. 2021b; Crago et al. 2021; McBride et al. 2019, 2020, 2021; Vijayakumar et al. 2019).

Our paper focused on this lack of inclusivity in the sex worker community, in research, laws, and different lobby groups, and on the need for greater representation of diversity in sex work advocacy and mobilization efforts. Herein, we sought to highlight the heterogeneity of this population on the basis of the perspectives of 10 members of SWOs from seven community organizations in Canada. Our study revealed the important role that SWOs play in not only facilitating access to health and social services but also providing spaces where people in sex work can gather in safe and supportive environments. Despite SWOs' efforts, sex workers' mobilization is still limited by micro-, meso-, and

macrolevel stigmatization that prevents and/or discourages some workers from accessing their programs, services, and mobilization efforts.

Study participants provided insights into the diversity in gender among people in sex work, the breadth of racial and Indigenous statuses, and the social statuses of people who use substances, live with disabilities, and experience housing precarity. Our findings highlighted how sex worker diversity varies geographically, which some participants attributed to unique histories of colonialism, migration, marginalization (e.g., racism, classism, etc.), and mobilization across the country. Embedded within these structural contexts were social and class-based tensions among clients of SWOs and how these tensions alerted SWO leadership to the needs of cis men, trans men, trans women, and nonbinary workers, many who have been overlooked in sex work research. Participants highlighted the spectrum of socioeconomic status among clients and organizational responses to the so called "whore-archy" and highlighted how this challenges mobilization efforts. The strategies of outreach, engagement, and compassionate inclusion shared by the participants can inform and enhance efforts to connect with groups of people in sex work (Oselin and Weitzer 2013; Gerassi et al. 2017).

Our participants shared that cultivating safe spaces of belonging for the full spectrum of sex workers was challenged by the day-to-day realities of their lives, including, for some, substance use; for others, a lack of access to personal hygiene resources/supplies; and for many, living in poverty. Despite these barriers, participants revealed a variety of strategies that their organizations used to enhance the diversity among their clientele and mobilize this diversity throughout communities of people in sex work. These include working with peers to improve outreach to hidden/under-represented populations (e.g., Black people, im/migrants, sex workers who use drugs, those who are precariously housed, etc.), learning what specific resources are needed to best support underserved groups (e.g., having committees, workshops, etc.), providing leadership opportunities and building client capabilities (e.g., mentorship, training events, etc.), improving cultural safety across organizational practices (e.g., territorial acknowledgments, intake forms, ceremony, etc.), and cultivating safe spaces of belonging (e.g., group guidelines; shared ethics/values; philosophies of compassion, empathy, and respect; etc.).

Mobilizing the spectrum of sex workers is challenging because of the structural environment that inevitably restricts SWO outreach (Belle-Isle et al. 2014; Benoit and Unsworth 2022). Our study participants discussed how engaging a diverse leadership team, representative of sex workers in the community, needed to be an equity-centered strategy to mobilize community members. This strategy also encouraged individual capacity building by supporting people to develop leadership skills. Similar to participants in our study, allied SWOs around the globe aim to galvanize support for people in sex work by mobilizing the diversity of social locations, lived experiences, and life stages across sex worker populations (Vijayakumar et al. 2019). Some strategies involve collaborating with supportive allies across sectors, such as youth-serving organizations or substance-use harm-reduction organizations, and drawing on a variety of networks and resources (Belle-Isle et al. 2014; Vijayakumar et al. 2019). Moreover, sharing resources helps to enhance educational opportunities to learn about labor rights, increasing access to the justice system, advocating against the criminalization of labor, protesting against labor reforms, and building capacity among politicians, medical personnel, and law enforcement on sex workers' occupational needs (Dziuban and Stevenson 2018; Fedorkó et al. 2021; Foley 2019; Sex Workers' Rights Advocacy Network 2015; West et al. 2021).

In summary, the diversity of the sex worker population described by participants paints a much broader picture of the industry than is reflected in much of the current sex work literature, governmental laws (i.e., criminal code) and legislation, and conservative antiprostitution lobby groups. SWOs that engage in both service and advocacy draw strength from the variety of lived and living experiences of their clients as a key strategy in mobilizing sex workers on their rights as it accesses the "hidden" and "underserved" corners of the population that continue to be under-represented in sex work research, policies,

and programs (Oselin and Weitzer 2013). Importantly, SWOs that identify criminalization as a key barrier to accessing labor and human rights also highlight the intersectional factors that help or hinder individual capabilities to access those rights (Benoit and Unsworth 2022; Nussbaum 2003; Oselin and Weitzer 2013). Though a crucial strategy, the decriminalization of sex work alone will not solve all the injustices of inequity and disadvantage faced by differently marginalized sex workers. However, the failure to address diversity within sex work is common in research and policymaking and even in SWOs themselves. SWOs are, however, attempting to reconcile issues within their organizations, and their observations are instructive for allied partners who engage with them.

Our study has several limitations. All the participants identified as cis women, and therefore, some of their views on non-cis women in sex work may be colored by an outsider lens. Further, 3 of the 10 people interviewed do not have sex work experience, which may have resulted in similar outsider perspectives. In addition, as the insights from participants are limited to their personal experiences and those of the clients who access the services of the SWOs, there remains a degree of "invisibility" and lack of representation among some people who sell sexual services but do not have access to the services offered by SWOs. Furthermore, there may be researcher bias given that the second author oversaw the distribution of resources, recruited the study participants, conducted the interviews, and co-analyzed the data with the first author. However, we hope the participatory research approach of the project, which was collaboratively established *prior to* the interviews' taking place, minimized the potential for bias (Benoit and Unsworth 2021). In addition, while the represented SWOs provide services to several thousand sex workers across Canada, other rural, northern, inner-city geographies and Internet-based and independent escort populations are under-represented, limiting the generalizability of the findings to other communities in Canada and beyond.

## 6. Conclusions

Despite the small number of interviews in our study, our research shows that leaders of SWOs who provide services to people in sex work report a diversity among their clients that can provide an exemplary for government policymakers and academics. The qualitative findings show that SWOs can operate as structural interventions beyond decriminalization that can improve equitable access to health and social services for sex workers despite the many ways that sex workers are excluded at the micro, meso, and macro levels. Sex work scholars and policymakers should engage with leaders of SWOs as stakeholders when conducting participatory, sex-worker-led studies or when studies are led by experiential advisory teams to attain better representation of the diversity of people who sell sexual services.

**Author Contributions:** A.M. co-analyzed the data with C.B., and co-wrote the manuscript. C.B. conceptualized the project, co-designed the study with community partners, oversaw the distribution of project resources, recruited the study participants, conducted the interviews, and co-analyzed the data and co-wrote the manuscript with A.M. All authors have read and agreed to the published version of the manuscript.

**Funding:** Pierre Elliott Trudeau Foundation (Grant No. 5689).

**Institutional Review Board Statement:** This project received institutional ethics approval from the University of Victoria (Protocol No. 18-237).

**Informed Consent Statement:** Informed consent was obtained from all individual participants included in the study.

**Data Availability Statement:** Please contact the authors about data availability.

**Conflicts of Interest:** The authors declare that they have no conflict of interest.

## Note

1    "Two-Spirit is a way for Two-Spirit communities to organize—in other words, a way to identify those individuals who embody diverse (or non-normative) sexualities, genders, and gender expressions and who are Indigenous to Turtle Island" (Devor and Haefele-Thomas 2019, p. 134).

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
