# Peer review of "Understanding the Diversity of People in Sex Work: Views from Leaders in Sex Worker Organizations"

_socsci, doi:10.3390/socsci12030191_

Round 1
Reviewer 1 Report (Previous Reviewer 1)
The authors responded to all the comments that were previously done, and the clarity of the paper has improved. Furthermore, the arguments were strengthened by additional references.
Author Response
The authors responded to all the comments that were previously done, and the clarity of the paper has improved. Furthermore, the arguments were strengthened by additional references.
We thank you for your initial comments to help strengthen our paper and appreciate your revision and approval of our revised paper.
Reviewer 2 Report (Previous Reviewer 2)
Overall
Thank you for your revisions. However, I do not feel that this manuscript is ready for publication in its current format, and it also remains unclear as to how this paper would be an appropriate fit for the current special issue. The objectives of this paper would be better suited for a different format, e.g., a literature review or commentary.
In this new draft, you make clearer the argument that Canada’s laws are based on a binary framing of sex workers as women and victims, and clients as perpetrators and men. This is a much more compelling argument than the current objectives. However, this is not carried out throughout the paper. This could be a stronger direction for the paper but would require an extensive rewrite.
Introduction
Thank you for incorporating my previous feedback to engage with “the whore-archy” discourse. However, I feel it inappropriate to be citing works from 1855 and 1991 to introduce a largely “insider” term. Sex workers have written about the whore-archy at length, with many contemporary examples to choose from. I think that if the authors engage with more recent text, your framing of the whore-archy will be much better informed.
https://slutever.com/sex-worker-tilly-lawless-interview/
https://streethooker.wordpress.com/2020/04/30/looking-up-at-the-whorearchy-from-the-bottom/
https://aninjusticemag.com/what-is-the-whorearchy-and-why-its-wrong-1efa654dcb22
Methods
Thank you for adding a Strengths and Limitations section. However, it is still necessary to note that your interviewees are all ciswomen at the top of the methods section.
In the methods section, you write that of the organizations you interview, "Some only cater to women-identified (cis and trans) sex workers". This is brought up again in the results, as well as the need for tailored services and group workshops. This appears as a contradiction to the central argument of this paper. I suggest you give space to discuss the reasons as to why some services are tailored, and why the same can be true for research.
Findings
In a few cases, you include quotes that contain potentially harmful and stigmatizing language. It is fine to edit quotes for clarity as well removing harmful language or ideas.
· “we need to keep it centered around women, and ah, trans women, you know, men who have sex worked as women.”
· “I can’t really say that we have super strong leadership from Black and migrant and Asian communities, because of—well, the isolation, and years of colonization, and our social culture. [T]hose aren’t currently really strong, healthy, vibrant communities in our province or in our city.”
· “We had about half sex workers and half HIV—people living with HIV—attend our workshop, and we had a discussion about inclusive language with both groups”.
Discussion and Conclusion
Your discussion section focuses on the efforts of sex work organizations, and how this supports the wellbeing of sex workers. These are important findings, but not directly linked to the objectives of the paper.
The conclusion that sex work organizations access the ‘hard to reach’ corners of the population that continue to be underrepresented in sex work research, policies, and programs, has not been well supported.
Your conclusions and recommendations remain too broad, and therefore you do not reach the objectives of the paper. This paper would greatly benefit from clear and specific recommendations as to how researchers can improve sampling methods, representation, and participatory engagement with diverse sex workers.
The conclusion that “Sex work scholars and policymakers should include leaders of sex worker organizations when conducting their studies”, could be reframed to “conducting participatory, sex worker-led studies, or lead by experiential advisory teams. There is no inherent merit in working with only leaders of organizations, as they may themselves lack diversity in lived experience.
Author Response
Thank you for your revisions. However, I do not feel that this manuscript is ready for publication in its current format, and it also remains unclear as to how this paper would be an appropriate fit for the current special issue. The objectives of this paper would be better suited for a different format, e.g., a literature review or commentary. In this new draft, you make clearer the argument that Canada’s laws are based on a binary framing of sex workers as women and victims, and clients as perpetrators and men. This is a much more compelling argument than the current objectives. However, this is not carried out throughout the paper. This could be a stronger direction for the paper but would require an extensive rewrite.
Thank you for your advice on how to further revise out paper to be more coherent and better fit the special issue. We have revised the frontend and other parts of the paper so that the research findings contributes to the literature on moving beyond the binary framing of sex workers as cis women and victims of sexual violence, and clients as cis men and perpetrators of sexual violence.
Introduction
Thank you for incorporating my previous feedback to engage with “the whore-archy” discourse. However, I feel it inappropriate to be citing works from 1855 and 1991 to introduce a largely “insider” term. Sex workers have written about the whore-archy at length, with many contemporary examples to choose from. I think that if the authors engage with more recent text, your framing of the whore-archy will be much better informed.
Thank you for your positive feedback. We have revised our discussion of the “whore-archy” discourse with more recent references to the academic literature.
Methods
Thank you for adding a Strengths and Limitations section. However, it is still necessary to note that your interviewees are all ciswomen at the top of the methods section.
We thank the reviewer for this observation and have added this detail to the top of section 3.2
In the methods section, you write that of the organizations you interview, "Some only cater to women-identified (cis and trans) sex workers". This is brought up again in the results, as well as the need for tailored services and group workshops. This appears as a contradiction to the central argument of this paper. I suggest you give space to discuss the reasons as to why some services are tailored, and why the same can be true for research.
We thank the reviewer for this comment and have provided additional background as to why this lack of diversity may exist within sex worker organizations and how this is in part connected to the organizational stratification that exists within the sex industry (i.e., the ‘whore-archy’).
Findings
In a few cases, you include quotes that contain potentially harmful and stigmatizing language. It is fine to edit quotes for clarity as well removing harmful language or ideas.
- “we need to keep it centered around women, and ah, trans women, you know, men who have sex worked as women.”
- “I can’t really say that we have super strong leadership from Black and migrant and Asian communities, because of—well, the isolation, and years of colonization, and our social culture. [T]hose aren’t currently really strong, healthy, vibrant communities in our province or in our city.”
- “We had about half sex workers and half HIV—people living with HIV—attend our workshop, and we had a discussion about inclusive language with both groups”.
We thank the reviewer for this suggestion and have adjusted the quotes to remove stigmatizing language.
Discussion and Conclusion
Your discussion section focuses on the efforts of sex work organizations, and how this supports the wellbeing of sex workers. These are important findings, but not directly linked to the objectives of the paper.
We thank the reviewer for this comment and have clarified that one of the objectives of the paper is to consider sex worker organizations as the structural intervention working to improve equitable access to health and protective services for sex workers and expand their rights to occupation health and safety.
The conclusion that sex work organizations access the ‘hard to reach’ corners of the population that continue to be underrepresented in sex work research, policies, and programs, has not been well supported.
We thank the reviewer for this comment and have clarified our position that sex worker organizations (SWOs) have valuable contributions to make in designing policies and programs that can assist ‘hard-to-reach’ populations that may not be well represented in the current research literature. SWOs do so by their outreach efforts to mobilize underrepresented and marginalized workers into more positions of leadership and also offering options to participate if workers are not ‘out’.
Your conclusions and recommendations remain too broad, and therefore you do not reach the objectives of the paper. This paper would greatly benefit from clear and specific recommendations as to how researchers can improve sampling methods, representation, and participatory engagement with diverse sex workers.
We thank the reviewer for this comment and have clarified our conclusion to state that SWOs can offer insights to researchers and policy makers that seek to improve the quality of life of sex workers, based on their experience working with the community and navigating challenges associated with societal and occupational stigma.
The conclusion that “Sex work scholars and policymakers should include leaders of sex worker organizations when conducting their studies”, could be reframed to “conducting participatory, sex worker-led studies, or lead by experiential advisory teams. There is no inherent merit in working with only leaders of organizations, as they may themselves lack diversity in lived experience
We thank the reviewer for this comment and have included this language to specify the type of efforts where sex worker organizations could provide meaningful input.
Reviewer 3 Report (New Reviewer)
As intersectionality and life-course framework are the latest tools in critical studies, the manuscript "mobilizing the diversity of people in sex-work" adds value to these currencies by foregrounding it in the field of sexual commerce. By doing so, broadly, it not-only pins down heterogeneity in the social locations of sex workers (age, indigeneity, disability, people of colour, immigrants, etc) but also cogently dislodges the stereotypical images of sex workers as violated bodies and trafficked victims. One of the strengths of the manuscript is that it has intensively observed sex-work industry and sex worker organisations which is being reflected in terms of concrete insights. Thus, in many instances, I could relate with my ethnographic works on sexual commerce and community mobilisation projects in Southern India, for instance, page 8, 9 and 10 notes about the internal tensions, conflicts and differences that pervade among the sex workers and its implications of mobilising sex workers, etc. It also provided a detailed account of the methodology and how research ethics has been followed. While broadly, arguing for decriminalization of sex work, this paper brings a nuance argument of addressing these interlocking forms of oppression and highlights the importance of "compassion, non-judgment, toleration, etc" as the strategies to overcome the tensions.
Having discussed about the contribution of this manuscript, let me focus on the weak areas so that the arguments can be strengthened.
1- Prabha Kotiswaran's book "Dangerous sex and Invisible Labour" can provide some theoritical clues about "governance feminism" and why a criminalisation approach through feminist language is dangerous. Importantly, Kotiswaran and Svati Shah argue that sexual commerce is not an independent industry but part of informal economy. In addition, Shah's work points out the presence of female, male and transmen in sex industry. Thus, these two works can be used for making solid arguments as intersectionality which started by Black-feminists, highlighted how formal and organised sector discriminates black-women by providing unfair wage. Same is seen is sexual commerce where income differences can be found among sex workers, it is also linked to the status of their customers, etc.
2- Since intersectionality is the lens through which this manuscript alarms the point of invisibility and lack of representation in community mobilisation, experience of sex workers who were not associated with sex worker organisation could have brought many nuanced aspects. However, as the manuscript shows many of field insights did not move beyond the account of clients or staffs who were part of sex worker organisations. Thus, if at one level, their narration indicated intra-group tensions, at another level, those tensions were also tackled, as if there are no areas which is beyond the reach/purview of these organisation, particularly the macro aspects like politico-economy of the state. Similarly, evidence of police violence against the sex workers from ethnic minority or indigenous groups compared to sex workers from privileged locations could have strengthened the arguments on intersectionality.
3- I think discussion section should carefully weave the arguments presented in the findings through field-insights instead of going beyond the findings.
Author Response
Comments and Suggestions for Authors
As intersectionality and life-course framework are the latest tools in critical studies, the manuscript "mobilizing the diversity of people in sex-work" adds value to these currencies by foregrounding it in the field of sexual commerce. By doing so, broadly, it not-only pins down heterogeneity in the social locations of sex workers (age, indigeneity, disability, people of colour, immigrants, etc) but also cogently dislodges the stereotypical images of sex workers as violated bodies and trafficked victims.
One of the strengths of the manuscript is that it has intensively observed sex-work industry and sex worker organisations which is being reflected in terms of concrete insights. Thus, in many instances, I could relate with my ethnographic works on sexual commerce and community mobilisation projects in Southern India, for instance, page 8, 9 and 10 notes about the internal tensions, conflicts and differences that pervade among the sex workers and its implications of mobilising sex workers, etc. It also provided a detailed account of the methodology and how research ethics has been followed. While broadly, arguing for decriminalization of sex work, this paper brings a nuance argument of addressing these interlocking forms of oppression and highlights the importance of "compassion, non-judgment, toleration, etc" as the strategies to overcome the tensions.
Having discussed about the contribution of this manuscript, let me focus on the weak areas so that the arguments can be strengthened.
1- Prabha Kotiswaran's book "Dangerous sex and Invisible Labour" can provide some theoretical clues about "governance feminism" and why a criminalisation approach through feminist language is dangerous. Importantly, Kotiswaran and Svati Shah argue that sexual commerce is not an independent industry but part of informal economy. In addition, Shah's work points out the presence of female, male and transmen in sex industry. Thus, these two works can be used for making solid arguments as intersectionality which started by Black-feminists, highlighted how formal and organised sector discriminates black-women by providing unfair wage. Same is seen is sexual commerce where income differences can be found among sex workers, it is also linked to the status of their customers, etc.
We thank the reviewer for this recommendation and have included this reference in our revised introduction and conceptual framework.
2- Since intersectionality is the lens through which this manuscript alarms the point of invisibility and lack of representation in community mobilisation, experience of sex workers who were not associated with sex worker organisation could have brought many nuanced aspects. However, as the manuscript shows many of field insights did not move beyond the account of clients or staffs who were part of sex worker organisations. Thus, if at one level, their narration indicated intra-group tensions, at another level, those tensions were also tackled, as if there are no areas which is beyond the reach/purview of these organisation, particularly the macro aspects like politico-economy of the state. Similarly, evidence of police violence against the sex workers from ethnic minority or indigenous groups compared to sex workers from privileged locations could have strengthened the arguments on intersectionality.
We thank the reviewer for this nuanced insight and have added this point as a limitation to our study findings.
3- I think discussion section should carefully weave the arguments presented in the findings through field-insights instead of going beyond the findings.
We thank the reviewer for this recommendation and have focused our discussion section on the findings identified through field insights.
This manuscript is a resubmission of an earlier submission. The following is a list of the peer review reports and author responses from that submission.
Round 1
Reviewer 1 Report
Mobilizing the Diversity of People in Sex Work: Views from Allies in Community Organization
This is an important manuscript as it incorporates an important discussion on the diversity of people engaging in sex work. However, it needs additional work to be suitable for publication.
Abstract
Much of the literature on people who sell sexual services presents them and their services in homogeneous categories, primarily ciswomen who sell street-based services. An unacceptable level of sampling bias is the outcome
What does unacceptable mean? Avoid adjectives in an academic manuscript. Also, what does much mean?
Maybe the extension of the literature on ciswomen is due to the number of ciswomen that engage in sex work therefore, there is a representative component explaining the number of studies that represent ciswomen. This should be discussed and considered when the main argument of this paper is sampling bias.
Introduction
When it is said “much” what does it mean? Provide contextualization, in the world? In Latin America? In some countries being trans or homosexual is still a crime…
The argument that there are no studies considering gender heterogeneity is poor as there is a myriad of studies considering other diversities that should be cited to strengthen such argument. Including men who have sex with men, trans women, etc.
https://www.ncbi.nlm.nih.gov/pmc/articles/PMC6565772/
https://pubmed.ncbi.nlm.nih.gov/24313294/
When it is mentioned that workplace diversity is often restricted to the street context, in which context this is referred? There is extensive literature that analyses other types of sex work venues an extended revision of the bibliography is suggested to strengthen this argument that it is not accurate. For example, review “Reimagining Sex Work Venues” by West et al. 2021
https://link.springer.com/chapter/10.1007/978-3-030-64171-9_12, West et al. 2022, Pitpitan et al. 2014 etc.
What is the difference between prostitution and sex work? This should be stated in the introduction and should not be used as synonyms.
One important issue of this manuscript it is that is not contextualized so it falls into generalizations that are not applicable in many regions of the world so a pre-bias in the arguments of such paper are embedded.
Materials and Methods
The main argument of the study is that there are not enough studies considering gender diversity in sex work studies but the participants are ciswomen. How is this not a limitation? If the interviews were only conducted in Canada why this study is not contextualized in that country?
Findings
Although the findings are interesting they do not necessarily respond to the main research question but more into providing a descriptive perspective of the diversity of sex work in Canada, creating safe spaces and mobilizing the diversity on levels. How this is contributing to the extension of diversity in the overall literature?
Reviewer 2 Report
Overall
· This paper raises important considerations in regard to representation in sex work research. This paper aims to discuss how sex work researchers may improve sampling methods, which could be a very valuable contribution to the literature. However, the current draft demonstrates a disconnect between aims and results and fails to reach its objectives.
· As this is a methods paper, it may be helpful for the authors to engage with other literature on methodology relating to sex work research, community-based/led research, or literature on sampling and engaging with marginalized communities.
· The paper would greatly benefit from an additional “Strengths and Limitations” section
Introduction
· On page 2, the authors write, “focusing on gender and/or marginalization alone limits the applicability of findings to the most marginalized of workers” and “studies[…]remain focused on recruiting the most marginalized workers and, in the absence of comparative data, risk skewing perceptions that sex workers as a whole, are subject to the same social and structural complexities”.
I appreciate the authors acknowledging that sex work is extremely diverse, and there is an ongoing need for more inclusive sampling approaches. However, based on recent literature and sex work community discourse, the above argument appears overly simplistic. The authors may wish to utilize an intersectional framework, which would allow for a more nuanced analysis of sex workers’ varying experiences of marginalization and relative privilege within the community. Sex work discourse around the “whorearchy” also speaks to this. Through this lens, we see the importance in centering the needs of multiply marginalized sex workers in research and policy recommendations, and how such recommendations, in turn, will also benefit more privileged sex workers. There remains an opportunity for this paper to more thoughtfully examine how sex work research can disrupt heterogeneous depictions of sex work communities while also acknowledging that multiply marginalized sex workers continue to be the most impacted by research, policy, policing and criminalization.
· The paper first discusses limitations in sampling methods and representation among existing sex work literature, and aims to address the need for greater representation of the diversity of sex workers in research. The authors indicate that the results draw from interviews with staff of sex work organizations. The justification for use of this data set/participant sample is not sufficient. It is recommended that the authors further elaborate on how this data set is best suited for the research objectives.
· 3.2. Interview procedure. The authors very briefly describe that the research sample includes 10 staff members of sex work organizations, all of whom are ciswomen. Firstly, this is a rather small sample for qualitative research. As well, given this paper aims to discuss diversity in sampling, it would be helpful to know more demographic information of your participants (any lived experience in sex work, racialization, etc.). If your sample lacks diversity, this should be discussed in a limitations section. If participant information needs to be protected for anonymity, that should be indicated.
Results
· The coding themes ”Creating safe spaces for diversity at the local level, and Mobilizing the diversity of sex workers at multiple levels” do not have a clear link to the research objectives. Can you further define what is meant by ‘local level’ and ‘multiple levels’, and how they may relate to considerations in sex work research?
· The subsections within the results end abruptly. These sections would benefit from concluding interpretations that connect back to the research objectives.
Discussion
· In the discussion, you begin to connect how strategies employed at sex work community organizations could be useful for improving sampling methods for sex work research. As this is the central aim of the paper, these connections could be further emphasized. The final theme, (second to last paragraph of the discussion section), does not have an applicable use in sex work research. Please ensure each theme is linked back to the research objective.
Conclusions
· The conclusions emphasize key takeaways on how sex work organizations mobilize diversity, however, there is no clear indication of how such strategies can be applied in a research setting. Conclusions/recommendations should be applicable to sex work research so that they are connected to the paper’s objectives.